# Antimicrobial Resistance and Comparative Genome Analysis of *Klebsiella pneumoniae* Strains Isolated in Egypt

**DOI:** 10.3390/microorganisms9091880

**Published:** 2021-09-05

**Authors:** Radwa Abdelwahab, Munirah M. Alhammadi, Ehsan A. Hassan, Entsar H. Ahmed, Nagla H. Abu-Faddan, Enas A. Daef, Stephen J. W. Busby, Douglas F. Browning

**Affiliations:** 1Institute of Microbiology and Infection, School of Biosciences, University of Birmingham, Birmingham B15 2TT, UK; radwa.wahab418@gmail.com (R.A.); MMA707@student.bham.ac.uk (M.M.A.); S.J.W.BUSBY@bham.ac.uk (S.J.W.B.); 2Faculty of Medicine, Assiut University, Assiut PO 7151, Egypt; dr_ehsan66@yahoo.com (E.A.H.); entsar.2012@yahoo.com (E.H.A.); nhi-af@hotmail.com (N.H.A.-F.); deafenas@yahoo.com (E.A.D.); 3Biology Department, Princess Nourah bint Abdulrahman University, Riyadh 11671, Saudi Arabia; 4College of Health & Life Sciences, Aston University, Aston Triangle, Birmingham B4 7ET, UK

**Keywords:** *Klebsiella pneumoniae*, antibiotic resistance, virulence, whole genome sequencing

## Abstract

*Klebsiella pneumoniae* is an important human pathogen in both developing and industrialised countries that can causes a variety of human infections, such as pneumonia, urinary tract infections and bacteremia. Like many Gram-negative bacteria, it is becoming resistant to many frontline antibiotics, such as carbapenem and cephalosporin antibiotics. In Egypt, *K. pneumoniae* is increasingly recognised as an emerging pathogen, with high levels of antibiotic resistance. However, few Egyptian *K. pneumoniae* strains have been sequenced and characterised. Hence, here, we present the genome sequence of a multidrug resistant *K. pneumoniae* strain, KPE16, which was isolated from a child in Assiut, Egypt. We report that it carries multiple antimicrobial resistance genes, including a *bla*_NDM-1_ carbapenemase and extended spectrum β-lactamase genes (i.e., *bla*_SHV-40_, *bla*_TEM-1B_, *bla*_OXA-9_ and *bla*_CTX-M-15_). By comparing this strain with other Egyptian isolates, we identified common plasmids, resistance genes and virulence determinants. Our analysis suggests that some of the resistance plasmids that we have identified are circulating in *K. pneumoniae* strains in Egypt, and are likely a source of antibiotic resistance throughout the world.

## 1. Introduction

*Klebsiella pneumoniae* is a Gram-negative, encapsulated, non-motile bacterium that can cause a variety of human infections, including pneumonia, urinary tract infections, bacteremia, and liver abscesses [1]. It readily colonises human mucosal surfaces, such as the gastrointestinal tract and oropharynx, where it is viewed as a commensal, but it is from these sites that it gains entry to other tissues to cause severe disease. In the past, *K. pneumoniae* infections have mostly targeted immunocompromised patients in hospital settings, but the emergence of hypervirulent (*hv) K. pneumoniae* strains has seen an increase of community-acquired infection in healthy individuals [1,2,3].

In *K. pneumoniae*, a number of virulence factors are important for human infection. These include the production of capsule and lipopolysaccharide (LPS), and the expression of fimbriae (both types I and III) and siderophore transport systems. For *hv K. pneumoniae* strains, the overproduction of capsule is important, due to the action of the plasmid encoded RmpA and/or RmpA2 transcription factors, and such stains usually possess multiple siderophore systems (e.g., aerobactin and yersiniabactin) as well as other iron transport systems (e.g., *kfuABC*) [1,2,3,4].

Like many bacteria, *K. pneumoniae* is now becoming resistant to many frontline antibiotics, limiting the treatment of both nosocomial and community acquired infections [1,2,3]. Two types of antibiotic resistance are particularly important, the acquisition of extended spectrum β-lactamase (ESBL) genes (e.g., *bla*_TEM_*, bla*_CTX-M_ and *bla*_OXA_), which provide resistance to cephalosporin and monobactam antibiotics, and genes encoding carbapenemases (e.g., *bla*_NDM_, *bla*_VIM_ and *bla*_KPC_), providing resistance to carbapenems [1,2,3]. In particular, carbapenemase producing *K. pneumoniae* have a higher incidence of mortality due to lack of treatment options [5]. Both ESBL and carbapenemases genes are often located on self-transmissible plasmids, contributing to their ease of spread [1,2,3].

In Egypt, *K. pneumoniae* is increasingly recognised as an emerging pathogen, showing high levels of antibiotic resistance [6,7,8,9,10,11,12,13]. Furthermore, documented transmission of carbapenem resistant *K. pneumoniae* from Egypt to other countries has been observed [14,15,16,17,18]. Despite this, the genomes of very few *K. pneumoniae* strains, isolated in Egypt, have been sequenced and fully characterized [19,20]. During a recent screen in Assiut, Egypt [21], we isolated a carbapenem resistant *K. pneumoniae* strain from a child with diarrhoea. Here we present the antimicrobial resistance profile and genome sequence of this strain, *K. pneumoniae* KPE16, and compare it with other Egyptian isolates, identifying common plasmids, resistance genes and virulence determinants. We demonstrate that Egyptian *K. pneumoniae* strains carry specific antimicrobial resistance plasmids, which are similar to those isolated from travellers who previously visited Egypt. We suggest that these plasmids are circulating in *K. pneumoniae* strains in Egypt and are possibly a source of worldwide antibiotic resistance transmission.

## 2. Materials and Methods

### 2.1. Isolation and Characterisation of Klebsiella pneumoniae Strain KPE16

*Klebsiella pneumoniae* strain KPE16 was isolated from a child, presenting with diarrhoea to the outpatients clinic of Assiut University Children’s Hospital in 2016 [21]. Isolation and original identification were carried out at the Medical Research Center, Faculty of Medicine, Assiut University with growth on MacConkey agar, Eosin Methylene Blue (EMB) agar and IMViC testing, suggesting that KPE16 was an *Escherichia coli* isolate. However, analysis of the draft genome sequence of KPE16 (see below) with SpeciesFinder 2.0 and KmerFinder 3.2 software at the Center for Genomic Epidemiology (CGE) (http://www.genomicepidemiology.org/, accessed on 9 April 2021) clearly indicated that KPE16 had been misclassified and was *K. pneumoniae* [22,23,24]. *K. pneumoniae* KPE16 was tested for susceptibility to a range of antimicrobial agents, using the Kirby-Bauer disc diffusion method [25], which was interpreted according to the CLSI 2014 [26]. The antimicrobial discs (Hi-Media, India) contained the following antibiotics: imipenem (10 µg), meropenem (10 µg), trimethoprim/sulfamethoxazole (5 µg), cefaclor (30 µg), ceftriaxone (30 µg), ampicillin (10 µg), ciprofloxacin (5 µg), oxytetracycline (30 µg), amoxicillin (25 µg), norfloxacin (10 µg), tobramycin (10 µg), and amikacin (30 µg).

### 2.2. Genome Sequencing of K. pneumoniae Strain KPE16

Complete genome sequencing of *K. pneumoniae* strain KPE16 was carried out using Illumina sequencing by Microbes NG (https://microbesng.com/, accessed on 9 April 2021). Plated cultures were inoculated into a cryopreservative (Microbank™, Pro-Lab Diagnostics, Birkenhead, UK). Ten to 20 µL of this suspension were lysed with 120 µL of TE (10 mM Tris, 1 mM EDTA, pH 7.5) buffer containing lysozyme (final concentration 0.1 mg mL^−1^) and RNase A (ITW Reagents, Barcelona, Spain) (final concentration 0.1 mg mL^−1^), incubated for 25 min at 37 °C. Proteinase K (VWR Chemicals, Cleveland, OH, USA) (final concentration 0.1 mg mL^−1^) and SDS (Sigma-Aldrich, St. Louis, MO, USA) (final concentration 0.5% *v*/*v*) were added and incubated for 5 min at 65 °C. Genomic DNA was purified using an equal volume of SPRI beads and resuspended in EB buffer (Qiagen, Hilden, Germany). DNA was quantified with the Quant-iT dsDNA HS kit (ThermoFisher Scientific, Rockford, IL, USA) assay in an Eppendorf AF2200 plate reader (Eppendof UK Ltd., Stevenage, UK). Genomic DNA libraries were prepared using the Nextera XT Library Prep Kit (Illumina, San Diego, CA, USA), following the manufacturer’s protocol with the following modifications: 2 ng of DNA were used as input, and PCR elongation time was increased to 1 min from 30 s. DNA quantification and library preparations were carried out on a Hamilton Microlab STAR automated liquid handling system (Hamilton Bonaduz AG, Bonaduz, Switzerland). Pooled libraries were quantified using the Kapa Biosystems Library Quantification Kit for Illumina on a Roche light cycler 96 qPCR machine. Libraries were sequenced with the Illumina HiSeq using a 250 bp paired end protocol. Illumina reads were adapter trimmed using Trimmomatic 0.30 with a sliding window quality cutoff of Q15 [27]. Genome assembly was performed using Unicycler v0.4.0 [28] and contigs were annotated using Prokka 1.11 [29]. This Whole Genome Shotgun project has been deposited at DDBJ/ENA/GenBank with the sequence data for *K. pneumoniae* KPE16 under the accession number JAGFBT000000000. The genome sequence (chromosome and plasmids) of *K. pneumoniae* reference strain MGH 78578 was obtained from BioProject PRJNA31 [30], whilst the GenBank accession numbers for the genomes of Egyptian *K. pneumoniae* strains SF, SK, HM, and SP were RXLV00000000, RXLX00000000, RXLW00000000, and RXLY00000000, respectively [19].

### 2.3. Bioinformatic Analysis of K. pneumoniae Genome Sequences

Draft genomes were visualised using Artemis [31], comparisons between *K. pneumoniae* genomes were examined using the CGView Server (http://stothard.afns.ualberta.ca/cgview_server/, accessed on 9 April 2021) [32], the Basic Local Alignment Search Tool (BLAST) at NCBI (https://blast.ncbi.nlm.nih.gov/Blast.cgi, accessed on 9 April 2021) and the Artemis Comparison Tool (ACT) [33]. Representations of genome organisation were drawn using the CGView Server [32] and ACT [33]. *K. pneumoniae* sequence types were determined using MLST 2.0 [34], plasmid replicons were detected using PlasmidFinder 2.1 [35], antibiotic resistance gene analysis used ResFinder 3.2 [36], and virulence gene analysis was performed using VirulenceFinder 2.0 [37] with the online software from CGE (http://www.genomicepidemiology.org/, accessed on 9 April 2021). In addition, capsule typing, heavy metal resistance genes, and virulence genes were identified using the BIGSdb-Kp database at the Institut Pasteur (https://bigsdb.web.pasteur.fr/index.html, accessed on 9 April 2021) and the Virulence Factor Database (http://www.mgc.ac.cn/VFs/main.htm, accessed on 9 April 2021) [38]. Capsule typing and O-antigen locus type were also determined using Kaptive Web (https://kaptive-web.erc.monash.edu/, accessed on 9 April 2021) [39], insertion sequences were identified using ISfinder (https://www-is.biotoul.fr/blast/resultat.php, accessed on 9 April 2021) [40], and bacteriophage were determined using PHASTER (https://phaster.ca/, accessed on 9 April 2021) [41].

## 3. Results

### 3.1. Antimicrobial Resistance and Genomic Characterisation of K. pneumoniae Strain KPE16

*K. pneumoniae* strain KPE16 was isolated from a child, at the outpatient clinic of Assiut University Children’s Hospital in 2016 [21]. As multidrug resistant *K. pneumoniae* are a major concern in Egypt [6,7,8,9,10,11,12,13], we examined its susceptibility to various antimicrobials. Results detailed in Table 1 show that *K. pneumoniae* strain KPE16 was resistant to all antibiotics tested, including the carbapenem antibiotics, imipenem and meropenem, and the cephalosporin antibiotics, cefaclor and ceftriaxone, suggesting that it possesses carbapenemase and ESBL genes.

To understand more about the antibiotic resistance and virulence genes that KPE16 carries, we sequenced its genome and compared it to the *K. pneumoniae* reference strain MGH 78578, and four other recently sequenced Egyptian isolates (*K. pneumoniae* strains SF, SK, HM and SP) which have not been fully characterised (Table 2) [19,30]. Our analysis indicated that out of all the *K. pneumoniae* strains examined, SP possessed the smallest genome, containing the fewest coding sequences, whilst the genomes of other strains were relatively similar in size and gene content (Table 2). Comparison of each strain’s genome with the MGH 78578 chromosome indicated that there were a number of common regions of difference (RODs) between the Egyptian strains and MGH 78578 (Figure 1). Although many RODs locate to genes that encode proteins of undefined function, differences between genes responsible for surface structures (e.g., fimbriae, an outer membrane porin, a type I secretion system (TISS), and capsule and O-antigen LPS biosynthetic operons) and prophage were detected. Consistent with this, each Egyptian strain differed from MGH 78578 in their capsule K type and O-antigen locus type: KPE16 was predicted as K43 O2v1, strains SF, SK, and HM were K17 O1v1, and SP was K36 O4 (Table 2). Note that none of the strains were capsule type K1 or K2, which is associated with *hv K. pneumoniae* [1,2,3]. The Egyptian *K. pneumoniae* strains also belong to diverse sequence types, with KPE16 belonging to sequence type ST1399, SF, SK, and HM were all ST101, and SP was ST3050 (Table 2). Norsigian et al. [20] suggested that *K. pneumoniae* strains SF, SK, and HM are very similar. This was confirmed by comparing the draft genome assemblies of each isolate, which showed that SK and HM were almost identical, and SF only differed from SK and HM due to the absence of a number of prophage genes and an IncFIB(pQil) plasmid (Table 2 and Appendix A).

### 3.2. Characterisation of Plasmids Carried by Egyptian K. pneumoniae Strains

Many *K. pneumoniae* strains carry multiple plasmids, for example, MGH 78578 carries five plasmids (Table 2) [2,3,30]. Consistent with this, analysis of each Egyptian strain identified a number of plasmid replicons (Table 2) [35]. The genome of *K. pneumoniae* strain KPE16 contains four IncF replicons and one IncQ1 replicon. Due to the draft nature of the KPE16 genome assembly, each plasmid replicon was on a separate contig and it is not clear how many plasmids this strain possesses. It is of note that MGH 78578 plasmids pKPN3 and pKPN4 both carry IncFIB and IncFII replicons (Table 2). This arrangement occurs on other IncF plasmids and, in spite of replicons belonging to the same plasmid incompatibility group, subtle differences in copy number control regions mean that plasmids can be still maintained in the same cell [21,35,42,43]. Analysis of the IncFII(K) replicon (as well as other contigs from the KPE16 genome assembly) indicated that *K. pneumoniae* strain KPE16 carries a plasmid very similar to plasmid p2 (Figure 2A and Appendix A). This plasmid was found in a *K. pneumoniae* strain that was isolated from the urine of an Egyptian man in the USA in 2012 [14]. Significantly, this plasmid carries multiple antibiotic resistance determinants, including a *bla*_NDM-1_ carbapenemase and ESBL genes, amongst others (e.g., *bla*_NDM-1_, *bla*_CTX-M-15_, bla_TEM-1A_, bla_OXA-9_, *aac*(6′)-Ib-cr, *aad*A1, *aph*(3′)-VI, *qnrS1*) (Figure 2) [14]. Comparison of the KPE16 genome with the Egyptian strains SF, SK, HM, and SP identified one relatively large contig (contig 19) that was only present in KPE16 (Appendix A). BLAST analysis indicated that this region was carried by plasmid pKP1–19, which was isolated from an environmental *K. pneumoniae* strain KP-1 in Australia and carries heavy metal resistance genes for arsenic, silver, and copper (Figure 2B and Appendix A) [44]. The organisation of the KPE16 IncQ1 replicon was also found to be similar to the small broad-host range plasmid pRSF1010, which carries *sul2*, *aph*(3″)-Ib and *aph*(6)-Id resistance genes [35,45] (Appendix A). Thus, it likely that *K. pneumoniae* strain KPE16 carries three plasmids, a small IncQ1 plasmid, and two larger plasmids, each of which carry two IncF replicons.

The *K. pneumoniae* strains SK and HM carry three distinct plasmid replicons, whilst strain SF carries two, lacking the InFI(pQil) replicon (Table 2 and Appendix A). Analysis of the IncC and IncM1 replicons carried by SF, SK, and HM indicated that they were identical to those carried by plasmids pSH111_166 and pKpn14-4, respectively (Figure 3A,B). Plasmid pSH111_166 was found in a multi drug resistant *Salmonella enterica* subsp. enterica serovar Heidelberg strain that was isolated from a bovine diagnostic specimen in Ohio, USA in 2001 and carries a number of antibiotic resistance genes (i.e., *aac*(3)-IV, *bla*_CMY-2_, *aph*(4)-Ia, *aph*(6)-Id, *aph*(3″)-Ib, *floR, sul2, tetA*) (Figure 3A) [46]. Plasmid pKpn14-4 was carried by *K. pneumoniae* strain Kpn-14 isolated from a patient in Ontario, Canada in 2014, who had previously been hospitalised in Egypt and also possesses multiple antibiotic resistance genes (i.e., *aph*(3′)-VIb, *aph*(3″)-Ib, *aph*(6)-Id, *bla*_CTX-M-14b_, *qnrS1*) (Figure 3B) [15]. Analysis of the InFI(pQil) replicon from both SK and HM, indicated that they were identical to that of MGH 78578 plasmid pKPN4. Genome subtraction, as well as comparison of pKPN4 with the complete genomes of SK and HM, indicated that each strain carries a similar plasmid, though it lacks the *tra* genes of pKPN4 and is likely much smaller (Figure 3C). It is of note that pKPN4 also carries a multiple antimicrobial resistance genes (*aac*(6′)-Ib, *aac*(6′)-Ib-cr, *aad*A1, *bla*_OXA-9_ψ, *bla*_SHV-12_, *bla*_TEM-1A_) (Table 3 and Figure 3C).

The *K. pneumoniae* isolate SP also contained three different plasmid replicons (Table 2), suggesting that it possesses three plasmids. Analysis of the IncFIB(K) replicon indicated it was 99.8% identical to plasmid pTHO-004-1, which was isolated from *K. pneumoniae* strain THO-004, infecting a patient in Tokyo, Japan in 2018 (Figure 4). Plasmid pTHO-004-1 carries many gene associated with heavy metal resistance, such as arsenic, copper, and silver (Figure 4). The contig carrying the IncR replicon was similar to that carried by a number of IncR plasmids, though similarity between these plasmids was limited to the replicon region only (not shown). Interestingly, the IncQ1 replicon in *K. pneumoniae* strain SP is also very similar to that carried by *K. pneumoniae* strain KPE16 and pRSF1010 [35,45] (Appendix A).

### 3.3. Antibiotic- and Heavy Metal-Resistance Genes Carried by Egyptian K. pneumoniae Isolates

In the genome of *K. pneumoniae* KPE16, we detected various antibiotic resistance genes (Table 3). These can result in aminoglycoside resistance (*aac*(6′)-Ib-cr, *aac*(6′)-Ib, *aadA1*, *aph*(6)-Id, *aph*(3″)-Ib, *aph*(3′)-Ia), β-lactam resistance (*bla*_SHV-40_, *bla*_TEM-1B_, *bla*_OXA-9_*, bla*_NDM-1_, *bla*_CTX-M-15_), fosfomycin resistance (*fosA5*), quinolone/fluoroquinolone resistance (*aac*(6′)-Ib-cr, *oqxA, oqxB, qnrS1*), sulphonamide resistance (*sul2*), and tetracycline resistance (*tetA*) [36]. This is consistent with the multidrug resistance phenotype of *K. pneumoniae* KPE16 (Table 1). In particular, its resistance to carbapenem antibiotics is due to the carriage of *bla*_NDM-1_ carbapenemase gene. Note that the *bla*_NDM-1_ gene is located on same contig as the IncFII(K) replicon (Appendix A) and that the *sul2, aph(3″)-Ib, aph(6)-Id* and *aph(3′)-Ia* genes co-localise with the IncQ1 replicon (Appendix A), indicating that these resistance determinants are plasmid-borne. Furthermore, the *aadA1, aac*(6′)-Ib, *bla*_CTX-M-15_ and *qnrS1* resistance genes are all localised on a large contig, which is similar to the sequence of *K. pneumoniae* plasmid p2, suggesting that these genes are likely plasmid encoded (Figure 2 and Appendix A).

Reference strain MGH 78578 also carries multiple antibiotic resistance genes that could offer resistance to aminoglycoside, β-lactam, fluoroquinolone, fosfomycin, phenicol, sulphonamide, and tetracycline antibiotics (Table 3) [36]. Many of these resistance genes are located on plasmids pKPN4 and pKPN5 (Table 3). As pKPN4 carries many *tra* genes it is potentially conjugative (Figure 3). Norsigian et al. [20] demonstrated that *K. pneumoniae* strains SF, SK, and HM were resistant to many classes of antibiotics (e.g., aminoglycosides, β-lactams (penicillins, cephalosporins, and carbapenems), fluoroquinolones, phenicols, polymixins, sulphonamides, and tetracyclines), whilst strain SP was generally less resistant. This is consistent with the antibiotic resistance gene profile observed for the four strains (Table 3 and Appendix A). Strains SF, SK, and HM all contained the *bla*_VIM-29_ carbapenemase gene, which explains their resistance to carbapenems, and all four Egyptian strains carry multiple ESBL genes. For SF, SK, and HM the *bla*_CTX-M-14b_ gene was co-localised with the IncM1 contig, indicating that it is plasmid borne, which is consistent with this plasmid being similar to pKpn14-4 (Figure 3B and Appendix A). For *K. pneumoniae* strain SP the *sul2, aph*(3″)-Ib, *aph*(6)-Id resistance genes co-localised with the IncQ1 replicon as previously mentioned (Appendix A).

As well as acquired antibiotic resistance genes, all five strains also possess a number of chromosomal point mutations in the genes encoding the outer membrane porins, *ompK36* and *ompK37*, which are associated with cephalosporin and carbapenem resistance, and the regulator *acrR*, which is associated with fluoroquinolone resistance (Table 3 and Appendix A) [36]. These, mutations could contribute to the antimicrobial resistance of each strain [36] and may explain why strain SP is resistant to the carbapenem antibiotic imipenem, yet seems to lack carbapenemase genes [3,20,47]. Isolates SF and SK were previously shown to be resistant to the polymixin antibiotic colistin, whilst strains HM and SP were sensitive [19,20]. Genome analysis of SF and SK indicates that products of the *phoPQ*, *pmrAB*, *mgrB*, *yciM,* and *lpxM* genes, which have all been implicated in colistin resistance [3,48,49,50,51,52], were either identical to the colistin sensitive reference strain MGH 78578 or differences were also present in HM and/or SP (Appendix A). Thus, the mechanism of colistin resistance in strains SF and SK is unclear.

In addition to antibiotic resistance genes, all strains carried loci for heavy metal resistance (Table 3). For example, *K. pneumoniae* KPE16 carries resistance genes for arsenic (*arsRDABC*), cobalt (*corA, corC, mgtA),* copper (*pcoABCDERS*), mercury (*merRTPCADE*), and silver (*silBCEFGPRS*). The arsenic, copper, and silver resistance determinates were all located on contig 19, which is also carried by plasmid pKP1-19 from the environmental *K. pneumoniae* strain, KP-1, (see above) (Figure 2B and Appendix A) [44], indicating that these genes are likely plasmid encoded. Due to the draft nature of the *K. pneumoniae* SF, SK, HM, and SP genome assemblies, it is unclear if their heavy metal resistance genes are plasmid borne. However, as such genes are often carried by plasmids (as seen for MGH 78578 pKPN3 and pKPN4 (Table 3)), it is likely that this will be the case, especially as we predict that SP contains a similar plasmid to pTHO-004-1 (Figure 4).

### 3.4. Virulence Genes of K. pneumoniae Isolates

Analysis of potential virulence determinants in each *K. pneumoniae* isolate (Table 3) indicated that all strains possess genes encoding the *E. coli* common pilus (*ecpRABCDE*), mannose-sensitive type I fimbriae (*fimABCDFGH*), and *Klebsiella* mannose-insensitive type III fimbriae (*mrkABCDF*). They also carry genes involved in biofilm (*treC* and *sugE*) and capsule (*rcsA* and *rcsB*) formation, as well as genes encoding urease (*ureDABCEFG*) [2,3,53]. With respect to siderophore production, each strain carried the genes for enterobactin production (*entAB* and *fepC*) and the aerobactin transporter (*iutA*) but not the aerobactin gene cluster (*iucABCD*) [1,2,3]. Only SK, SF, and HM carried the yersiniabactin uptake system (*irp1*, *ipr2*, *fyuA*) and the ferric uptake genes *kfuABC,* which are both associated with *hv K. pneumoniae* (Table 3) [1,2,3,4]. Thus, it is clear that each Egyptian isolate possess many genes associated with *K. pneumoniae* virulence. Note that none of the strains carried the RmpA or RmpA2 transcription factors, often found in *hv K. pneumoniae* strains [1,2,3].

## 4. Discussion

Here, we have reported the genome sequence of a multidrug-resistant *K. pneumoniae* strain, KPE16, isolated from an Egyptian child with diarrhoea. By comparing its genome with other Egyptian isolates, we have gained important insights into the distribution and mobility of virulence and resistance determinants, especially those that are plasmid-associated. As *K. pneumoniae* is a normal resident in the human gastrointestinal tract, we think it unlikely that strain KPE16 was the cause of diarrhoea [2,3]. However, irrespective of this, KPE16, alongside the other Egyptian *K. pneumoniae* isolates, possesses many genes associated with virulence, including those involved in capsule, siderophore, and fimbriae biosynthesis [1,2,3]. Many of these genes are shared with the archetype strain, MGH 78578, which is considered to be a low virulence strain (Table 3) [54]. Usually, *hv K. pneumoniae* strains have capsule type K1 or K2, carry the regulator RmpA/RmpA2 and have multiple siderophore genes and iron acquisition systems. None of the Egyptian strains fit this profile (Table 2 and Table 3), though strains SF, SK, and HM do possess an additional yersiniabactin uptake system and the *kfuABC* ferric transport genes [1,2,3,4,55,56]. Although, this may make SF, SK, and HM more virulent, as iron scavenging in important during infection [1,2,3,56], it is unlikely that any of these strains are hypervirulent in character.

KPE16 was resistant to all antibiotics tested (Table 1), highlighting the possibility that, if treatment were required with frontline antibiotics, it would likely fail to resolve an infection. This is consistent with the antibiotic resistance genes carried by this strain (Table 3). Of note is the presence of the *bla*_NDM-1_ carbapenemase and multiple ESBL genes (*bla*_SHV-40_, *bla*_TEM-1B_, *bla*_OXA-9_ and *bla*_CTX-M-15_), which likely account for this strain’s resistance to carbapenem and cephalosporin antibiotics (Table 1 and Table 3). *K. pneumoniae* strain KPE16 is sequence type ST1399, which has previously been detected in Egypt and was found, in each case, to be multidrug resistant [9,13]. Similarly, *K. pneumoniae* strains SF, SK, and HM were all multidrug resistant, possessing a plethora of antibiotic resistance genes, in particular, the *bla*_VIM-29_ carbapenemase gene and multiple ESBL genes. All three isolates were ST101, which is considered to be the most prevalent sequence type in Egypt and a major MDR sequence type throughout the world [3,9,13,57].

Like many *K. pneumoniae* strains, the Egyptian isolates presented here all carry plasmids, which encode antibiotic and/or heavy metal resistance determinants [3]. Our analysis indicates that *K. pneumoniae* strains KPE16 and SP carry an IncQ1 replicon, which is similar to that of pRSF100 [35,45] (Appendix A). This plasmid was also detected in Enteroaggregative *E. coli* strain E36, which we isolated in Assuit, Egypt in 2016 [21] and this suggests that similar IncQ1 plasmids are likely endemic in Egyptian Enterobacteriaceae (Appendix A). Furthermore, strain KPE16 possessed a plasmid very similar to *K. pneumoniae* plasmid p2, which was isolated in the USA from a man from Egypt [14] (Figure 2a). For *K. pneumoniae* strains SF, SK, and HM, we detected a plasmid similar to pKpn14-4 (Figure 2), which was isolated from a patient in Ontario, Canada, who had been previously hospitalised in Egypt. As such plasmids have been detected in Egyptian *K. pneumoniae* strains, as well as being isolated from patients that have visited Egypt, we suggest that these plasmids may be a source of antimicrobial resistance transmission worldwide. These data are consistent with the view that strains carrying high levels of resistance to antibiotics are not hypervirulent, and vice versa, presumably, because there is a trade-off, due the burden of carrying different genetic determinants. Notwithstanding this, the emerging acquisition of multidrug resistant by *hv K. pneumoniae* is particularly concerning [1,2,3,55,58,59,60].

## 5. Conclusions

In this study we used whole genome sequencing to determine the antimicrobial resistance genes and virulence determinants carried by a multi-drug resistant *K. pneumoniae* strain isolated in Assiut, Egypt, comparing it to other sequenced Egyptian isolates. We show that all *K. pneumoniae* strains examined carry multiple antimicrobial resistance genes, which include carbapenemase and/or ESBL genes, and that these determinants are likely plasmid encoded. Our analysis also identifies similar resistance plasmids carried by *K. pneumoniae* strains isolated from individuals who had visited Egypt.

*K. pneumoniae* has been defined as an ESKAPE pathogen (*Enterococcus faecium*, *Staphylococcus aureus, K. pneumoniae, Acinetobacter baumanii, Pseudomonas aeruginosa*, and *Enterobacter* species) and a particular global threat due to its increased resistance to antimicrobial agents [61]. It is clear that the success of *K. pneumoniae* as a human pathogen, in both healthcare and community settings, is due to the acquisition of antimicrobial resistance genes and specific virulence determinates on transmissible plasmids [1,2,3]. Thus, understanding the transmission of these plasmids and the population dynamics of *K. pneumoniae* is key in devising future strategies to intervene and disarm this important human pathogen.

## Figures and Tables

**Figure 1 microorganisms-09-01880-f001:**
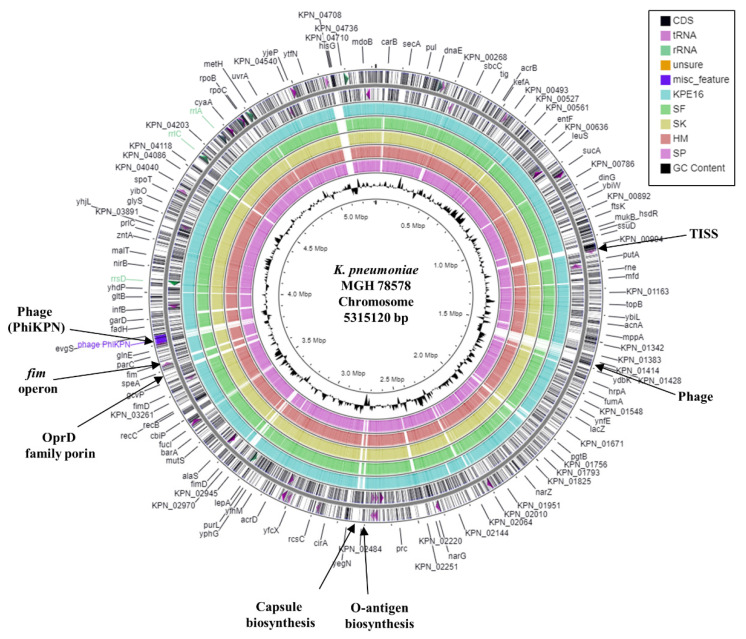
Comparison of the *K. pneumoniae* MGH 78578 chromosome with the draft genomes of *K. pneumoniae* strains KPE16, SF, SK, HM, and SP. The figure shows the comparison of the *K. pneumoniae* MGH 78578 chromosome with the genomes of *K. pneumoniae* strains KPE16, SF, SK, HM, and SP using GCview [32]. The outer two rings display the genes (CDS) and features of the *K. pneumoniae* MGH 78578 chromosome (CP000647.1) on both strands, with selected features labelled [30]. The blue, green, gold, red, and pink rings illustrate the BLAST results when the genome sequences of KPE16, SF, SK, HM, and SP, respectively, are compared to the MGH 78578 chromosome, with shaded regions indicating homology. The inner ring displays the GC content (black) for the *K. pneumoniae* MGH 78578 chromosome.

**Figure 2 microorganisms-09-01880-f002:**
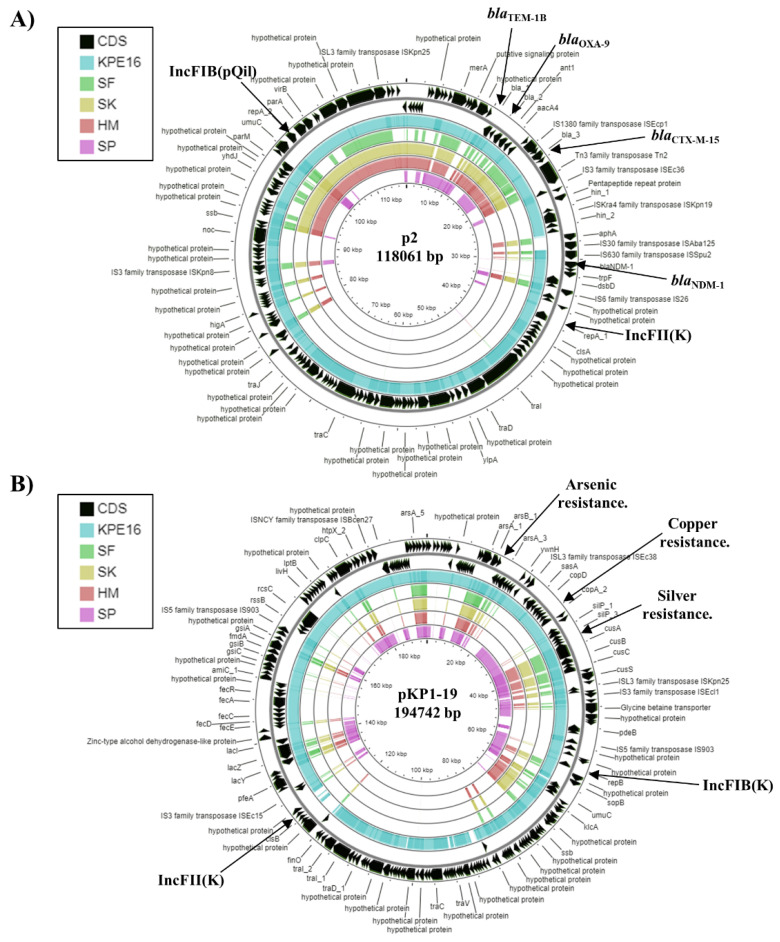
Analysis of plasmids carried by *K. pneumoniae* strain KPE16. The figure shows the comparison of the *K. pneumoniae* plasmids (**A**) p2 (CP009115.1) [14] and (**B**) pKP1–19 (CP012884.1) [44] with the genomes of *K. pneumoniae* strains KPE16, SF, SK, HM, and SP using GCview [32]. In both panels, the outer two rings display the genes (CDS) of each plasmid on both strands, with plasmid replicons and selected features labelled. The blue, green, gold, red, and pink shading illustrates the BLAST results when the genome sequences of KPE16, SF, SK, HM, and SP, respectively, are compared with each plasmid. Shaded regions indicate homology.

**Figure 3 microorganisms-09-01880-f003:**
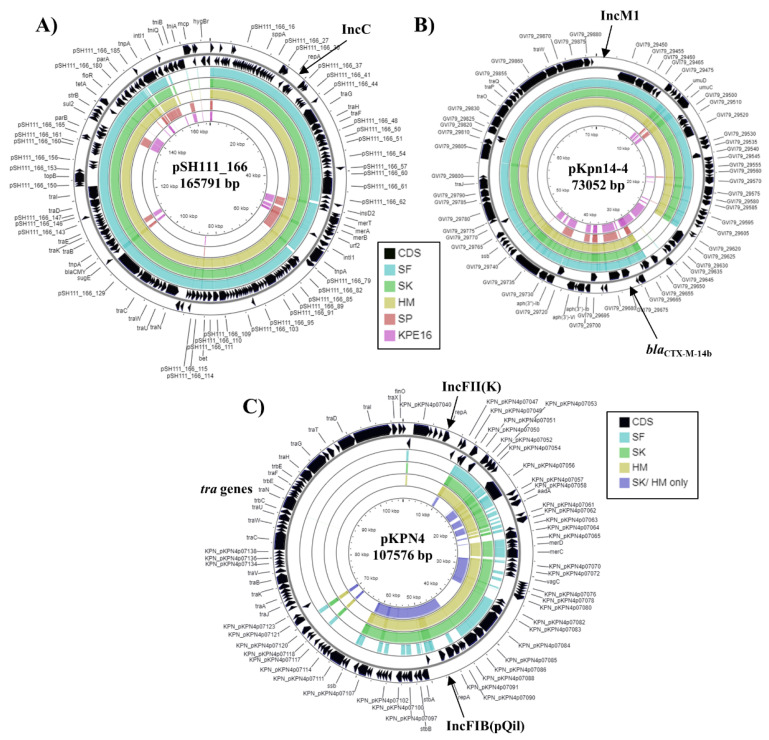
Analysis of plasmids carried by *K. pneumoniae* strains SF, SK, and HM. The figure shows the comparison of (**A**) *Salmonella enterica* subsp. enterica serovar Heidelberg plasmid pSH111_166 (JN983043.1) [46], (**B**) *K. pneumoniae* plasmid pKpn14-4 (CP047704.1) [15] and (**C**) *K. pneumoniae* MGH 78578 plasmid pKPN4 (CP000649.1) [30] with the genomes of *K. pneumoniae* strains SF, SK, HM, SP and KPE16, using GCview [32]. In all panels, the outer two rings display the genes (CDS) of each plasmid on both strands, with plasmid replicons labelled. In panels (**A**,**B**) the blue, green, gold, red, and pink shading illustrates the BLAST results when the genome sequences of SF, SK, HM, SP, and KPE16, respectively, are compared with each plasmid. In panel (**C**), the light blue, green, gold, and dark blue shading illustrates the BLAST results when the genome sequences of SF, SK, HM and the contigs that are specific to SK/HM only (after genome subtraction) are compared to pKPN4. Shaded regions indicate homology.

**Figure 4 microorganisms-09-01880-f004:**
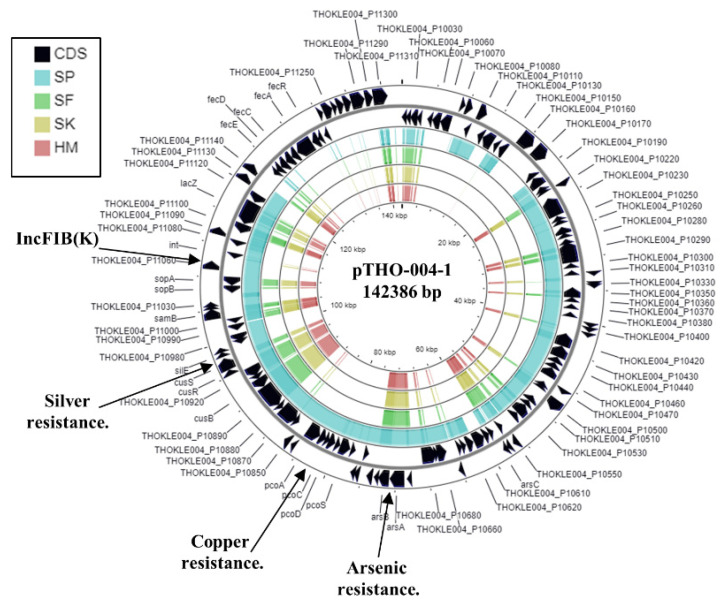
Analysis of the IncFIB(K) replicon carried by *K. pneumoniae* strain SP. The figure shows the comparison of *K. pneumonia* plasmid pTHO-004-1 (AP022528.1) with the genomes of *K. pneumoniae* strains SP, SF, SK, and HM, using GCview [32]. The outer two rings display the genes (CDS) on both strands, with the IncFIB(K) plasmid replicon and selected features labelled. The blue, green, gold, and red shading illustrates the BLAST results when the genome sequences of SP, SF, SK, and HM, respectively, are compared with pTHO-004-1. Shaded regions indicate homology.

**Table 1 microorganisms-09-01880-t001:** Antimicrobial resistance profile of *K. pneumoniae* strain KPE16.

Antibiotic	Antibiotic Class	Phenotype ^a^	Antibiotic	Antibiotic Class	Phenotype ^a^
Amoxicillin	penicillin	R	Amikacin	aminoglycoside	R
Ampicillin	penicillin	R	Tobramycin	aminoglycoside	R
Cefaclor	cephalosporin	R	Ciprofloxacin	fluoroquinolone	R
Ceftriaxone	cephalosporin	R	Norfloxacin	fluoroquinolone	R
Imipenem	carbapenem	R	Oxytetracycline	tetracycline	R
Meropenem	carbapenem	R	Trimethoprim/Sulfamethoxazole	antifolate/sulfonamide	R

^a^*K. pneumoniae* KPE16 was tested for susceptibility to a range of antimicrobial agents, using the Kirby-Bauer disc diffusion method [25], which was interpreted according to the CLSI 2014 [26]. Antibiotic resistant phenotype is designated R.

**Table 2 microorganisms-09-01880-t002:** Genome analysis of *Klebsiella pneumoniae* strains isolated in Egypt.

	MGH 78578	KPE16	SF	SK	HM	SP
Isolation details	1994	Assiut, Egypt. 2016	Cairo, Egypt. 2012	Cairo, Egypt. 2012	Cairo, Egypt. 2012	Cairo, Egypt. 2012
Genome size	5,694,894 bp	5,767,326 bp	5,684,559 bp	5,762,428 bp	5,759,337 bp	5,343,887 bp
G/C % ^a^	57.48%	56.93%	56.89%	56.85%	56.86%	57.44%
Genes (CDS)	5185	5452	5396	5459	5459	4974
Sequence type	ST38	ST1399	ST101	ST101	ST101	ST3050
Capsule type	K52	K43	K17	K17	K17	K36
O-antigen type	OL101	O2v1	O1v1	O1v1	O1v1	O4
Plasmid replicons	**pKPN3:** IncFIB, IncFII **pKPN4:** IncFIB(pQil),IncFII **pKPN5:** IncR **pKPN6:** Col440I **pKPN7**: untypeable	IncFIB, IncFIB(pKPHS1), IncFIB(pQil), IncFII(K), IncQ1	IncC, IncM1	IncC, IncFIB(pQil), IncM1	IncC, IncFIB(pQil), IncM1	IncFIB, IncQ1, IncR

^a^ G/C % of the bacterial chromosome is given for *K. pneumoniae* strain MGH 78578 [30].

**Table 3 microorganisms-09-01880-t003:** Analysis of antimicrobial resistance, heavy metal resistance, and virulence genes carried by Egyptian *K. pneumoniae* strains.

	MGH78578	KPE16	SK	SP
**Acquired antimicrobial resistance genes**	**Chromosome:***bla*_SHV-40_, *fosA, oqxA, oqxB* **pKPN4:***aac*(6′)-Ib, *aac*(6′)-Ib-cr, *aad*A1, *bla*_OXA-9_ψ ^a^, *bla*_SHV-12_, *bla*_TEM-1A_ **pKPN5:** *ant*(2″)-Ia, *aph*(3′)-Ia, *aph*(3″)-Ib, *aph*(6)-Id, *bla*_TEM-1B_, *catA1*, *cmlA1*, *sul1*, *sul2*, *tetD*	*aac*(6′)-Ib-cr, *aac*(6′)-Ib, *aadA1*, *aph*(6)-Id, *aph*(3″)-Ib, *aph*(3′)-Ia, *bla*_SHV-40_, *bla*_TEM-1B_, *bla*_OXA-9_, ***bla*****_NDM-1_**, *bla*_CTX-M-15_ *fosA5, oqxA, oqxB, qnrS1, sul2, tet(A)*	*aac*(6′)-Il, *aac*(6′)-Ib-cr, *aac*(6′)-Ib, *aadA22, aadA23, aph*(3″)-Ib, *aph*(3′)-VIb, *aph*(6)-Id, *bla*_SHV-28_, ***bla*****_VIM-29_**, *bla*_CMY-4_, *bla*_CTX-M-14b_, *bla*_OXA-9_ *dfrA1, floR, fosA, oqxA, oqxB, sul2, tet(A)*	*aph*(3′)-Ia, *aph*(3″)-Ib, *aph*(6)-Id, *bla*_SHV-28_*, bla*_CTX-M-15_, *bla*_TEM-1B_ *dfrA14, fosA, mph(A), oqxA, oqxB, sul2*
**Chromosomal point mutations associated with antimicrobial resistance**	***ompK36***: N49S, L59V, G189T, F198Y, F207Y, A217S, T222L, D223G, E232R, N304E ***ompK37***: I70M, I128M, N230G ***acrR*:** P161R, G164A, F172S, R173G, L195V, F197I, K201M	***ompK36*****:** N49S, L59V, G189T, F198Y, F207Y, A217S, T222L, D223G, E232R, N304E ***ompK37*****:** I70M, I128M ***acrR***: P161R, G164A, F172S, R173G, L195V, F197I, K201M	***ompK36*****:** N49S, L59V, L191S, F207W, A217S, N218H, D224E, L228V, E232R, T254S ***ompK37***: I70M, I128M ***acrR:*** P161R, G164A, F172S, R173G, L195V, F197I, K201M	***ompK36:*** N49S, L59V, T184P ***ompK37*:** I70M, I128M ***acrR*:** P161R, G164A, F172S, R173G, L195V, F197I, K201M
**Heavy metal resistance** **genes**	**Chromosome:** *corA, corC, mgtA,* **pKPN3:** *arsRDABC, pcoABCDERS, silBCEFGPRS* **pKPN4:** *merACDE*	*arsRDABC, corA, corC, mgtA, merRTPCADE, pcoABCDERS, silBCEFGPRS*	*arsRDABC, corA, corC, mgtA*, *merRTPABDE*	*arsRDABC, corA, corC, mgtA, merRTPCADE, pcoACERS, silBCEFGPRS*
**Virulence genes**	*entA, entB, ecpRABCDE, fepC, fimABCDFGH, iutA, mrkABCDF, rcsA, rcsB, treC, sugE, ureDABCEFG*	*entA entB, ecpRABCDE, fepC, fimABCDFGH, iutA, mrkABCDF, rcsA, rcsB, traT, treC, sugE, ureDABCEFG*	*entA, entB, ecpRABCDE, fepC, fimABCDFGH, fyuA, irp1, irp2, iutA, kfuABC, mrkABCDF, rcsA, rcsB, treC, sugE, ureDABCEFG*	*entA, entB, ecpRABCDE, fepC, fimABCDFGH, iutA, mrkABCDF, rcsA, rcsB, treC, sugE, ureDABCEFG*

^a^ ψ pseudogene.

## Data Availability

This Whole Genome Shotgun project has been deposited at DDBJ/ENA/GenBank with the sequence data for *K. pneumoniae* KPE16 under the accession number JAGFBT000000000.

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
