# Peer review of "Antimicrobial Resistance and Comparative Genome Analysis of Klebsiella pneumoniae Strains Isolated in Egypt"

_microorganisms, 2021, doi:10.3390/microorganisms9091880_

Round 1
Reviewer 1 Report
This work is a comprehensive and systematic study of “Antimicrobial resistance and comparative genome analysis of Klebsiella pneumoniae strains isolated in Egypt.”
It was very interesting to read the complete manuscript. After reading and thoroughly analyzing the data (text, figures, and tables) presented by the authors, I have decided to recommend a minor revision of the manuscript.
The conclusion appeared to be misinformative, and I suggest authors revise the following sentences.
“Our analysis also identifies similar resistance plasmids carried by K. pneumoniae strains isolated from individuals who had visited Egypt, suggesting transmission of these plasmids to different countries around the globe”.
So please be careful while postulating and this may become politically controversial.
Author Response
Authors’ comment: We thank the Reviewer for their positive assessment of our manuscript and positive comments. As requested we have modified the sentence in the Conclusion section (line 375), to only state “Our analysis also identifies similar resistance plasmids carried by K. pneumoniae strains isolated from individuals who had visited Egypt“, and have omitted the potentially controversial portion of the sentence “suggesting transmission of these plasmids to different countries around the globe”. We hope that this is satisfactory.
Reviewer 2 Report
The paper is well written and can be accept in present form.
Author Response
Authors’ comment: We thank the Reviewer for their positive assessment.
This manuscript is a resubmission of an earlier submission. The following is a list of the peer review reports and author responses from that submission.
Round 1
Reviewer 1 Report
The manuscript titled "Antimicrobial resistance and comparative genome analysis of Klebsiella pneumoniae strains isolated in Egypt" has focused on the multidrug-resistant K. pneumoniae strain, KPE16, isolated from a child in Assiut, Egypt. The title reflects the actual contents and primary analysis of the various carbapenemase and extended-spectrum β-lactamase genes. The most recent and updated citations well support the manuscript. The article certainly deserves a place for publication with few minor suggestions.
Authors may provide future challenges and conclusions separately.
Reviewer 2 Report
The paper report suggests that some of the resistance plasmids that we have identified are circulating in K. pneumoniae strains in Egypt, and are a source of
antibiotic resistance throughout the world.
The paper is well written but I have some comments:
- the Materials and Methods are poor: improve. Add the method used for identification.
- Discussions add the following references:
- Fasciana et al. Draft genome sequence and biofilm production of a carbapenemase-producing Klebsiella pneumoniae (KpR405) sequence type 405 strain isolated in Italy. Antibiotics 2021.
-
Co-existence of virulence factors and antibiotic resistance in new Klebsiella pneumoniae clones emerging in south of Italy. BMC Infect Dis. 2019
Add the conclusion section.
Reviewer 3 Report
In this manuscript, the authors sequenced an multi-drug resistant strain of K. pneumoniae strain, KPE16, and used the full genome to perform a comparative genomic analysis [using other Egyptian isolates].
This reviewer is not agree with the conclusion - Based on the plasmid sequences of five strains it would be inaccurate to say that "These plasmids are circulating in K. pneumoniae strains in Egypt and are a source of worldwide antibiotic resistance transmission. In order to support this conclusion, the authors need to perform the detail evolutionary analysis on a big dataset.